# **Evolution of NO<sub>x</sub> in the Denver Urban Plume during the Front Range Air Pollution and Photochemistry Experiment**

Carlena J. Ebben<sup>1</sup>, Tamara L. Sparks<sup>1</sup>, Paul J. Wooldridge<sup>1</sup>, Teresa L. Campos<sup>2</sup>, Christopher A. Cantrell<sup>3</sup>, Roy L. Mauldin<sup>3</sup>, Andrew J. Weinheimer<sup>2</sup>, Ronald C. Cohen<sup>1,4</sup>

<sup>1</sup>Department of Chemistry, University of California Berkeley, Berkeley, California 94720, United States
 <sup>2</sup>National Center for Atmospheric Research, Boulder, Colorado 80301, United States
 <sup>3</sup>Department of Atmospheric and Oceanic Sciences, University of Colorado, Boulder, Colorado 80309, United States
 <sup>4</sup>Department of Earth and Planetary Sciences, University of California Berkeley, Berkeley, California 94720, United States

Correspondence to: Ronald C. Cohen (rccohen@berkeley.edu)

**Abstract.** As NO<sub>x</sub> (NO<sub>x</sub>=NO+NO<sub>2</sub>) is transported away from cities, it undergoes photochemical oxidation to peroxynitrates (RO<sub>2</sub>NO<sub>2</sub>,  $\Sigma$ PNs), alkyl nitrates (RONO<sub>2</sub>,  $\Sigma$ ANs), and nitric acid (HNO<sub>3</sub>). These higher oxide species each have different lifetimes to permanent removal or conversion back to NO<sub>x</sub>, resulting in nitrogen oxide chemistry that evolves as plumes are transported away from cities. Here, observations from the Front Range Air Pollution and Photochemistry Experiment

- (FRAPPÉ) are used to describe the evolution of NO<sub>x</sub> and NO<sub>y</sub> (NO<sub>y</sub>≡NO<sub>x</sub>+ΣPNs+ΣANs+HNO<sub>3</sub>+...) as the Denver urban plume flows outward from the city center. We evaluate the chemistry, dilution, and deposition rates in the plume to provide numerical constraints on the NO<sub>x</sub> and NO<sub>y,i</sub> lifetimes. We find that plume dilution with background air occurs with a lifetime of 3.5 hours. NO<sub>x</sub> concentrations decrease more rapidly with a lifetime to chemical loss and dilution of 2 hours in the near field of the city center. NO<sub>y</sub> has an effective lifetime of 3 hours and due to a combination of HNO<sub>3</sub> deposition and dilution.
- The results provide a useful test of conceptual and numerical models of chemistry during the evolution of urban plumes.

#### **1** Introduction

The chemistry of nitrogen oxides regulates tropospheric oxidants including hydroxyl radicals (OH), nitrate radicals (NO<sub>3</sub>), and ozone (O<sub>3</sub>). Nitrogen oxide chemistry also affects formation of secondary organic aerosols (SOA) both through regulation of oxidants and, as has been recently demonstrated, through formation and condensation of organic nitrates (Fry et

- 5 al., 2014; Ayres et al., 2015; Rollins et al., 2012; Lee et al., 2014). In urban areas, emissions of NO<sub>x</sub> (≡NO+NO<sub>2</sub>) result from combustion for transportation, power generation, home heating, and industrial processes (Dallmann and Harley, 2010). In more rural locations, microbial nitrification and denitrification in soils, biomass fires, and lightning are the dominant sources (Hudman et al., 2012; Mebust et al., 2011; Schumann and Huntrieser, 2007). During transport out of cities, urban plumes are subject to mixing with the free troposphere and horizontal expansion, effects that result in the physical dilution of the high
- 10 concentrations found in the urban core. At the same time, oxidation and deposition act on the individual chemical species in the plume, resulting in differential production and removal and setting the spatial pattern of nitrogen deposition. For the nitrogen oxides, chemical processes that occur during plume evolution lead to the formation of higher oxides, including peroxynitrates (RO<sub>2</sub>NO<sub>2</sub>, ΣPNs), alkyl and multifunctional nitrates (RONO<sub>2</sub>, ΣANs), and nitric acid (HNO<sub>3</sub>). During daytime the major pathways for NO<sub>x</sub> oxidization are the reactions:

| 15 | $NO_2 + OH + M \rightarrow HNO_3 + M$                  | (R1)  |
|----|--------------------------------------------------------|-------|
|    | $NO + RO_2 + M \rightarrow RONO_2 + M$                 | (R2a) |
|    | $NO + RO_2 \rightarrow RO + NO_2$                      | (R2b) |
|    | $NO_2 + R(O)O_2 + M \rightleftharpoons R(O)OONO_2 + M$ | (R3)  |

Understanding the balance between the various oxidation pathways of NO<sub>x</sub> and whether and how the higher oxides of NO<sub>x</sub>,
NO<sub>z</sub> (≡RO<sub>2</sub>NO<sub>2</sub>+RONO<sub>2</sub>+HNO<sub>3</sub>+...), are returned to NO<sub>x</sub> by further chemistry as a plume moves from the center of cities to more rural locations is essential to understanding ozone and aerosol. Here we use observations from the Front Range Air Pollution and Photochemistry Experiment (FRAPPÉ) that took place in the region around Denver, Colorado during summer 2014 to investigate spatial patterns of NO<sub>x</sub> oxidation and NO<sub>y,i</sub> production and deposition with the goal of describing the processes affecting evolution of NO<sub>x</sub> and NO<sub>y,i</sub> concentrations downwind of urban sources.

## 25 2 Observations

FRAPPÉ used the NCAR C-130 aircraft during July and August 2014 to sample air in the Northern Front Range region around Denver, Colorado. Flights centered in large part on the area immediately surrounding Denver to understand the regional chemistry and meteorology as it affects air quality. Research objectives also included assessing upslope flow into the foothills of the Rockies west of Boulder, CO and investigating agricultural and oil and gas emissions in the area north

30 and east of Denver.

 $NO_2$  and higher oxides of nitrogen were measured using thermal dissociation laser induced fluorescence (TDLIF) (Day et al., 2002). Briefly, a pulsed frequency-doubled YAG laser is used to pump a dye laser at 585 nm that provides excitation for the

detection of ambient NO<sub>2</sub> by laser-induced fluorescence. The fluorescence signal at wavelengths >700 nm is collected and imaged onto a red sensitive photocathode using gated photon counting techniques to discriminate against prompt scattering. The thermal dissociation portion of the instrument is operated using four channels: ambient temperature for detection of NO<sub>2</sub>,  $\sim$ 200° C for total RO<sub>2</sub>NO<sub>2</sub>,  $\sim$ 350° C for total RONO<sub>2</sub>, and the fourth channel held at  $\sim$ 550° C and used for HNO<sub>3</sub> (Day

- et al., 2002). By taking the difference between the NO<sub>2</sub> signals of two adjacent channels, the concentrations of each class of molecules are obtained. Calibration of the instrument was carried out by flowing mixtures of zero air and known concentrations of NO<sub>2</sub> standard gas. These measurements have previously been shown to compare well with other instruments when making similar measurements (Beaver et al., 2012; Thornton et al., 2000; Wooldridge et al., 2010). Other observations used in this analysis include NO measured by chemiluminescence (Weinheimer et al., 1994). VOCs were
- measured using the trace organic gas analyzer (TOGA) a fast GC/MS instrument (Apel et al., 2003; Apel et al., 2010). The TOGA instrument runs on a 2 minute sampling cycle, and all measurements used in this analysis are 2 minute averages on that time base. Measurements of CO were made by vacuum UV resonance fluorescence (Gerbig et al., 1999). HO<sub>x</sub> radicals, including OH, were measured by chemical ionization mass spectrometry (Mauldin et al., 2001; Hornbrook et al., 2011). We analyzed observations within the boundary layer (<1,000 m above ground level) and averaged over all wind speeds and</p>
- directions. Transport time was approximated as the transport distance divided by the median observed wind speed of 13 km hr<sup>-1</sup>.

## 3 The lifetimes of NO<sub>y,i</sub>

Denver is an isolated urban area. As an approximation, we think of it as an extended point source emitting into air with concentrations of trace chemicals characteristic of the continental background. In other examples downwind of a city, the

20 transition from urban emissions to background conditions typically occurs over length scales of order 50 km (Dillon et al., 2002; Perring et al., 2013). Here we describe observations of the spatial evolution of NO<sub>x</sub> and its higher oxides within and downwind of Denver and examine the role of physical and chemical factors that set the lifetime of nitrogen oxides in the Denver plume.

We begin with the simplest model of the NO<sub>x</sub> lifetime,  $\tau_{NOx}$ , where  $\tau_{NOx}$  is set by only two factors – dilution and reaction

- with OH. Dilution rates are typically reported in the range 0.1-0.25 hr<sup>-1</sup>, corresponding to a lifetime of 4–10 hours (Dillon et al., 2002; Lin et al., 1996; Nunnermacker et al., 1998). At 10<sup>7</sup> molec cm<sup>-3</sup> OH, the NO<sub>x</sub> lifetime to chemical removal by OH is comparable, ~4 hours. Combining this chemical removal with dilution, the NO<sub>x</sub> lifetime is predicted to be 2 hours. The lifetime of NO<sub>x</sub> with respect to midday oxidation by OH in other plumes has been reported to range from 2–8 hours (Alvarado et al., 2010; Valin et al., 2013; Ryerson et al., 1998; Ryerson et al., 2003). Additional terms in the atmosphere that
- affect the apparent chemical lifetime of  $NO_x$  include reactions to form organic nitrates ( $\Sigma PNs$  and  $\Sigma ANs$ ), reactions to produce  $NO_x$  from its higher oxides, and emission and deposition of both  $NO_x$  and the higher oxides downwind of the source region. Different treatment of these terms in previous analyses of  $NO_x$  lifetimes reported in the literature make it difficult to

effectively summarize the apparent lifetime of  $NO_x$  in other plumes or to compare the results here to those plumes. For example, the effect of temperature on both source molecules for peroxyacyl radicals and on the lifetime of peroxynitrates can impact the apparent lifetime of  $NO_x$ . Additionally, peroxynitrates can serve to extend the  $NO_x$  lifetime (LaFranchi et al., 2009; Finlayson-Pitts and Pitts Jr., 1999). In the southeast U.S. during summer where it is warm and isoprene emissions are

5 strong, Romer et al. (2016) showed that the sum of  $NO_x$  and peroxynitrates function as a family where the family lifetime is set by chemical conversion to  $RONO_2$  and  $HNO_3$ , and the  $\Sigma PNs:NO_x$  ratio is maintained in steady-state at a near constant value. As we show below, circumstances in the Denver plume are different, and  $\Sigma PNs$  are effectively a permanent sink of  $NO_x$ .

Figure 1a shows the concentrations of NO<sub>x</sub> (gray),  $\Sigma$ PNs (blue),  $\Sigma$ ANs (green), and HNO<sub>3</sub> (red) during transport away from

- Denver. NO<sub>x</sub> is the most abundant NO<sub>y</sub> component near the city, and its concentration quickly drops off, due to a combination of dilution and oxidation. Consistent with our simple model, the observed NO<sub>x</sub> lifetime, defined as the point where the city center value decreases to e<sup>-1</sup>, is ~2 hours. Figure 1b shows the fractional composition of NO<sub>y</sub>, as a function of distance from Denver. In the area directly surrounding Denver, NO<sub>x</sub> is unsurprisingly the largest fraction of NO<sub>y</sub>, ~50%. HNO<sub>3</sub> initially is ~25% of NO<sub>y</sub>, and ΣPNs and ΣANs account for ~10% and ~15%, respectively. As air is transported away
- from Denver and undergoes chemical processing, the NO<sub>x</sub> fraction decreases, falling to ~20% of NO<sub>y</sub>. The fractions of ΣPNs and ΣANs each increase to approximately 20% of NO<sub>y</sub>. HNO<sub>3</sub> becomes the dominant fraction, making up ~40% of NO<sub>y</sub> after 100 km of transport. These fractions are typical of previous observations (Farmer et al., 2011; Browne and Cohen, 2012; Romer et al., 2016). The total concentration of NO<sub>y</sub> decreases by a factor of 4 over the first 160 km of transport out of Denver, due to dilution and deposition.
- In order to isolate the role of dilution in plume evolution, we compare the NO<sub>y,i</sub> chemicals to CO, which is a chemically conserved tracer on the time scale of plume evolution with a lifetime to chemical removal of  $\tau = 3$  days in full sunlight, corresponding to ~1% hr<sup>-1</sup> at 10<sup>7</sup> molec cm<sup>-3</sup> OH. We assume CO is diluted by mixing with background air and derive a dilution lifetime subject to that constraint. The loss of all other species to dilution is assumed to occur at the same mixing rate with different values for the background concentration. To facilitate a comparison to CO, it is useful to define a
- fractional difference F:

$$F = (X - X_{background}) / (X_{initial} - X_{background})$$
<sup>(1)</sup>

which normalizes the decay of molecules observed in the plume to the difference between their initial and final concentrations (Perring et al., 2010). By definition, F begins at 1 and ends at 0.

For a molecule lost only to dilution, the concentration would decay at the same rate as that of CO resulting in a fractional difference identical to that of CO at every time and location along the plume. Species with fractional difference lower than that of CO are, in the net, being removed by chemistry or deposition, while those with fractional difference greater than that of CO are being produced within the plume. Figure 2 shows the fractional difference, F, of each NO<sub>y</sub> species and of CO over the first 160 km of transport out of Denver. Initial concentrations, *X*<sub>initial</sub>, are defined as the median values observed within ~10 km of Denver. The background conditions, *X*<sub>background</sub>, are defined as the median values observed at 160 km from

Denver. The campaign average initial and background concentrations are 4290 and 475 ppt for  $NO_x$ , 835 and 422 ppt for  $\Sigma PNs$ , 1270 and 439 ppt for  $\Sigma ANs$ , 2240 and 1200 ppt for  $HNO_3$ , 9.2 and 2.5 ppb for  $NO_y$  and 150 and 91 ppb for CO. Adjusting the initial and background concentrations within the standard variance of the mean or median does not change our interpretation of the data.

- 5 Using the mean wind speed of 13 km hr<sup>-1</sup>, the transport time to reach the e<sup>-1</sup> point for CO is 3.5 hours, corresponding to a dilution rate of approximately 0.29 hr<sup>-1</sup>. This dilution rate is similar to mixing timescales observed in previous studies in Denver and other locations. For example, McDuffie et al. (2016) derived a dilution rate of 0.38 hr<sup>-1</sup> from model fits of diel average observations at the Boulder Atmospheric Observatory, a location within the domain of our study, Dillon et al. (2002) derived a dilution rate of 0.2-0.22 hr<sup>-1</sup> was
- estimated by Nunnermacker et al. (1998) for the Nashville urban plume, and a dilution rate of 0.1 hr<sup>-1</sup> was estimated by Lin et al. (1996) for the Toronto urban plume.

Both NO<sub>y</sub> and NO<sub>x</sub> decrease to e<sup>-1</sup> more quickly than CO, in ~3 and ~2 hours, respectively. To highlight differences from effects of dilution on the NO<sub>y</sub> species, we compare the fractional difference ratio of each NO<sub>y</sub> species to that of CO, given by  $F_{X}/F_{CO}$ , in Fig. 3.

- For the first two hours of plume evolution, dilution dominates, and all of the molecules exhibit rapid decay. NO<sub>x</sub> decreases most rapidly, with a lifetime of ~2 hours. HNO<sub>3</sub> decreases most slowly, decaying by only 25% at 2 hours transport time. This HNO<sub>3</sub> decay is much slower than CO, which has decreased by 40% at 2 hours, an indication that HNO<sub>3</sub> is being produced faster than it is being deposited early in plume transport. Later in the plume, at 100 km distance, HNO<sub>3</sub> has a lower *F* than CO, indicating deposition has become more important than production. After the initial decay due to dilution,  $\Sigma$ PNs sustain
- the highest fractional difference, reflecting the chemical production of ΣPNs later in the plume at a rate that exceeds the rate of dilution. For the first ~3 hours, the *F* for ΣANs decreases more rapidly than CO, indicating ΣANs are lost to reaction with OH or deposition more rapidly than they are produced. After ~3 hours, ΣANs appear to be lost at approximately the same rate as CO.

As total NOy is conserved in chemical reactions, the observed loss of NOy beyond that of CO is a measure of the role of

- deposition. Using the difference between the e-folding lifetimes of CO and NO<sub>y</sub> 3.5 and 2.8 hours, respectively we derive a lifetime to deposition of NO<sub>y</sub> of ~14 hours. This corresponds to a deposition rate of 0.07 hr<sup>-1</sup>, assuming a boundary layer height of 1 km. Vertical profiles of NO<sub>2</sub> and other gases are consistent with a 1 km PBL estimate. If we assume HNO<sub>3</sub> is the only species depositing, then we derive  $V_{dep}$  for HNO<sub>3</sub> of 2 cm s<sup>-1</sup>. Neuman et al. (2009) reported the lifetime of HNO<sub>3</sub> to dry deposition was 14 hours in Houston. Nunnermacker et al. (2000) calculated an NO<sub>z</sub> lifetime in the Nashville urban plume of
- 8.7 hours, although they acknowledge that this short lifetime would correspond to an unexpectedly fast deposition velocity of 6.7 cm s<sup>-1</sup> for HNO<sub>3</sub>. In a pine forest environment, Farmer and Cohen (2008) calculated a deposition velocity of HNO<sub>3</sub> of 3.4 cm s<sup>-1</sup>. Nguyen et al. (2015) measured a deposition velocity for HNO<sub>3</sub> of  $3.8 \pm 1.3$  cm s<sup>-1</sup> based on eddy covariance observations at a rural site in Alabama. Our derived deposition velocity is lower than some of the recent measurements in forests, consistent with terrain that has less tree cover and therefore less surface roughness (Seinfeld and Pandis, 2006).

(2)

To a first approximation, the difference between F of CO and NO<sub>x</sub> represents loss due to chemistry, specifically reaction with OH and/or production of organic nitrates. If the primary oxidation pathway of NO<sub>2</sub> is reaction with OH to form HNO<sub>3</sub> ( $k_{OH+NO2} = 9.2 \times 10^{-12}$  cm<sup>3</sup> molec<sup>-1</sup> s<sup>-1</sup> at 298 K), then the difference between the lifetimes of NO<sub>x</sub> and CO can be used to calculate a lifetime with respect to reaction with OH of 5.3 hours. Assuming a ratio of NO<sub>2</sub> to NO<sub>x</sub> of 0.72, the derived OH

concentration is 7.9x10<sup>6</sup> molec cm<sup>-3</sup>. The OH concentration can also be estimated by assuming HNO<sub>3</sub> is in steady state and setting the production of HNO<sub>3</sub> via reaction of NO<sub>2</sub> with OH equal to its loss to dilution and deposition. Following Day et al. (2009) we incorporate the derived deposition and dilution rates, along with observed HNO<sub>3</sub> and NO<sub>2</sub> concentrations at several points along the transect of the plume, and solve for the OH concentration, according to Eq. (2):

 $k_{OH+NO_2}[NO_2]_t[OH] = K_{dep}[HNO_3]_t + K_{dil}([HNO_3]_t - [HNO_3]_{bg}),$ 

- where  $K_{dep}$  is the deposition rate (0.07 hr<sup>-1</sup>), equal to the deposition velocity divided by the boundary layer height (1 km), and  $K_{dil}$  is the dilution rate (0.29 hr<sup>-1</sup>). We note that applying the steady-state approximation to species with lifetimes on the timescale of hours where steady-state is not achieved could lead to over- or under-estimation of OH concentration, especially in the near field of large emissions sources (Fried et al., 2011). This calculation yields an OH concentration in the range  $4x10^6$  to  $1x10^7$  molec cm<sup>-3</sup>. The average OH concentration used in the calculation is  $\sim 7x10^6$  molec cm<sup>-3</sup>, similar to that
- derived from the observed lifetime of NO<sub>x</sub> and nearly identical to the observed median OH level during FRAPPÉ of  $7.3 \times 10^6$ molec cm<sup>-3</sup>. Although we use production and loss of HNO<sub>3</sub> in the calculation of OH, this analysis also implicitly includes loss to organic nitrates, since we use the observed NO<sub>x</sub> concentration as the source term for HNO<sub>3</sub>, which decreases due to oxidation by all sink species. The NO<sub>x</sub> lifetime with respect to photochemical oxidation can also be derived from the correlation between the NO<sub>x</sub> concentration and the calculated production rates of its oxidation products, i.e. the sum of 20 production of  $\Sigma$ PNs,  $\Sigma$ ANs, and HNO<sub>3</sub>. We find that this lifetime is ~3 hours.

Depending on conditions,  $\Sigma$ PNs may be appropriately considered as part of a short-lived reactive nitrogen pool (Romer et al., 2016) or a terminal sink of NO<sub>x</sub>. Production of peroxynitrates is initiated by reaction of oxygenated VOCs. Their primary loss mechanism, aside from dilution, is via thermal decomposition, followed by reaction of the peroxy radical with NO. Assuming steady state conditions, we are able to use these known loss mechanisms to calculate the production rate

of peroxynitrates:

$$P(\Sigma PNs)_t = L(\Sigma PNs)_{TD,t} + K_{dil} ([\Sigma PNs]_t - [\Sigma PNs]_{bg})$$
(3)

Using this method to solve for the production rate of  $\Sigma$ PNs, we determine that P( $\Sigma$ PNs) decreases from an average of ~400 ppt hr<sup>-1</sup> in Denver to ~100 ppt hr<sup>-1</sup> after 160 km of transport. P( $\Sigma$ PNs) can also be calculated directly from observations of the oxygenated VOCs that react to produce peroxy acyl radicals and the associated reaction rate constants, following the

30 derivation described by LaFranchi et al. (2009). This calculation gives a similar range of production rates throughout transport of the plume. We determine that the average lifetime of  $\Sigma$ PNs is ~4 hours, sufficiently long that  $\Sigma$ PNs can be treated approximately as separate from NO<sub>x</sub>, which has a lifetime of ~2 hours.

Production and loss of  $\Sigma$ ANs can be treated similarly.  $\Sigma$ ANs are produced by the reaction of NO with RO<sub>2</sub> radicals. This reaction pathway can also result in the production of NO<sub>2</sub> + RO radicals; the branching ratio associated with production of  $\Sigma$ ANs from these competing reactions is referred to as  $\alpha$ . P( $\Sigma$ ANs) is calculated using the summation of VOC reactivities and associated  $\alpha$  values. P( $\Sigma$ ANs) is highest in Denver, ~75 ppt hr<sup>-1</sup>, and the rate decreases quickly during the first 30 km of

5 transport, averaging about 10 ppt hr<sup>-1</sup> beyond 30 km. Based on these production rates, we determine that  $\tau(\Sigma ANs)$  is at least 9 hours, significantly longer than the lifetime of NO<sub>x</sub> or HNO<sub>3</sub>.

#### **4** Conclusions

The observations and analysis of plume evolution described here contribute to our overall understanding of air quality in the region around Denver. We find that most of the  $NO_x$  emitted in the Denver urban plume is deposited as  $HNO_3$  within 100 km

- of the city. Export to further regions is much smaller, and at 140 km the export is a small perhaps imperceptible increment above the background. At this distance, the  $NO_{y,i}$  distribution is 40% HNO<sub>3</sub> with equal amounts  $NO_x$ ,  $\Sigma PNs$ , and  $\Sigma ANs$ . In the intermediate distances between these endpoints, we note there is substantial net removal of  $NO_x$  and net production of PNs throughout the full extent of the plume. In contrast, in the near field of the city, we observe net production of HNO<sub>3</sub> that exceeds deposition, while beyond 100 km, the net sink to deposition exceeds the chemical product.
- We observe a dilution rate of 0.29 hr<sup>-1</sup>, and we find that over the first 2 hours of transport of the plume away from Denver, dilution is the primary loss process. The decay of NO<sub>y</sub> to e<sup>-1</sup> occurs in ~3 hours, as a result of dilution and the additional removal by deposition at a rate of 0.07 hr<sup>-1</sup>. This corresponds to a deposition velocity for HNO<sub>3</sub> of 2 cm s<sup>-1</sup>, a number on the low side of recent observations, presumably because of flatter terrain. NO<sub>x</sub> is lost most quickly from the plume, with  $\tau_{NOx} \approx 2$ hours, which can be derived using either observations of the decay of NO<sub>x</sub> concentration or fractional difference. The total
- $NO_y$  concentration decreases by a factor of 3–4 during transport, due to dilution and deposition, and we observe that the fractional composition of  $NO_y$  evolves throughout transport. HNO<sub>3</sub> has the largest increase as a fraction of  $NO_y$  during transport.

Data Availability. All data is available for download at the FRAPPÉ data archive (http://catalog.eol.ucar.edu/frappe)

Competing Interests. The authors declare that they have no conflict of interest.

Acknowledgements. We acknowledge funding support from the Regional Air Quality Council and the Colorado Department of Public Health and Environment, NASA (DISCOVER-AQ NNX10AR36G), and NSF (AGS 1352972 and DGE 1106400).

We thank the entire FRAPPÉ science team for support on this project.

15

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
