# Peer review of "Evolution of NOx in the Denver Urban Plume during the Front Range Air Pollution and Photochemistry Experiment"

_Atmospheric Chemistry and Physics, 2017_

## Referee Comment (RC1) · Anonymous Referee #1 · 24 Sep 2017

General comments:

This paper describes an analysis of aircraft data for speciated reactive nitrogen during the FRAPPE campaign in 2014. The analysis uses measurements of mixing ratios and partitioning among these compounds as a function of downwind transport from Denver to derive rates for production and loss in each class. The analysis is a potentially valuable contribution to the literature that compares observed nitrogen production and loss rates with what is known from previous studies and from idealized model calculations. It will therefore be of substantial interest to the readers of ACP.

The analysis does require clarification of several issues prior to publication. The major

concern is that there is insufficient description of the methodology used to derive the nitrogen partitioning downwind of Denver, Colorado, and no attempt to estimate either variability or error bars. Both are required for the reader to understand how the analysis was done, and for the statements to be placed on a quantitative footing rather than the rather qualitative description given in the text of the paper. The authors should include a full error or variability analysis, including clear error bars in all figures and a description of how they were derived.

Other issues are outlined further in the comments below.

Specific comments:

Page 4, line 9-10. It would be helpful to get a sense for the underlying data that is used to define "transport away from Denver" and Figure 1. Presumably the points in the figure are an average from sets of transect? How many data points per transect? How uniform is the wind field used to define the transport? How is distance defined (city center, downwind edge) etc. Are the data for a single day or an average of several flights? Which flight or flights? What sort of meteorology (clear sky, warm / cold, etc.)? A map or time series illustrating how this is done would be useful.

Figure 1, y-axis. Units are listed as ppt, which is a mixing ratio, but the label is concentration. Make label agree with units. The figure would also be easier to interpret if scaled from zero rather than the data minimum.

Page 4, line 12. The 1/e time does not need to be given as an approximate number, but could be fit as an exact number. How good is such a fit? Can it be overlaid in figure 1?

Page 4, lines 13-14 (and in following paragraph). Give exact numbers here as well, or else numbers with error bars if they can be defined, rather than approximation signs. If the 1 sig. fig. numbers given are correct, take away the approx. signs and indicate the precision to 1 sig. fig.

Figure 2. Same comment as above. Can the fit leading to the derived dilution rate be shown together with the data? How many parameters are required for this fit? Does one need an offset on the exponential, or can this term be set to zero. $F = A*exp(-k*x)$ vs $F = A*exp(-k*x) + F0$ ?

Figure 3. Possibly related to comment above. Why do the points all come back together at the end point? There does not seem to be any physical reason why they should be constrained together at 160 km transport distance, unless there is a constraint imposed. Is the assumption that all points are reaching their background value at 160 km? If so, is that realistic? Background concentrations should be defined from the upwind side of the urban area, not the far downwind limit of the transects, or else well outside of the urban plume.

Page 5, line 27. What is the variability in the 1 km boundary layer depth? This must not be a single number, as it should vary with time of day. What range, and subsequent uncertainty, does this give for NOy deposition?

Page 5, line 34. Not all readers will be familiar with the terrain and land cover near Denver. Some further description of the land cover together with any appropriate references will be useful to the audience for this paper.

Page 6, line 11-13. The potential error in the steady state approximation is more severe than stated here. Steady state is valid after an induction time equal to 3-4 times the inverse of the first order loss rate constant for the species in question (here HNO3). See for example, Pilling and Seakins, Reaction Kinetics. With an NOy lifetime (defined by HNO3) of 14 hours, the steady state induction would be significantly longer than a day. Is the analysis still valid? Can an uncertainty or correction to the steady state approximation be estimated?

Page 6, line 15. Is there a reference for the quoted OH concentration? Where does this come from? Aircraft or ground based measurements?

[Figure]

Page 6, lines 18-20. NOx lifetime of 3 hours is quoted here, but it's not clear how this was done from the description. Is this really derived from the sum of production rates of the oxidation products, or is this just a re-statement of the fit of the NOx decay after correction for dilution? Methodology not well stated in deriving this number.

Page 6, line 24. How good should one expect the steady state to be for peroxy nitrates?

Page 6, line 30. "The method described by LaFranchi (2009)" should be provided explicitly. Which oxygenates are used in this calculation? Acetaldehyde only, or other aldehydes? What are the mixing ratios of these species, and do they evolve in a manner that is consistent with the model? Is a detailed model chemistry that includes production and loss of oxygenates included? If so, how detailed?

Page 6, line 31. It is not clear where the average lifetime of 4 hours comes from. Is this from equation (3), and if so, which variables are used here (e.g., the equation was used a few sentences earlier to determine production, assuming the loss was known?). Is the lifetime from the decay analysis presented earlier? Is it from the calculation in the preceding comment? What is the loss process of the peroxy nitrates? Is it reaction of the acyl peroxy radicals after thermal dissociation, or is it some other process?

Also, why does a 4 hour lifetime differentiate peroxy nitrates from NOx, which has a 2 hour lifetime? These time scales are not that different. Further justification needed.

Page 7, line 10. "Perhaps imperceptible increment" is a qualitative statement that would be easily quantified. How large is the increment, and what is the error bar or variability? See comments above. This is the major need of this paper to describe how much data is in each data point and what sort of variability in incorporated into the averages.
* * *

---

## Referee Comment (RC2) · Anonymous Referee #2 · 4 Oct 2017

The manuscript by Ebben et al. describes the photochemical evolution of NOx in the Denver urban plume, taking into account photochemistry, dilution and dry deposition. It thus addresses an important topic, investigation NOx transformation and loss in the continental boundary layer. The topic itself is highly relevant and thus suitable for ACP. Unfortunately, I feel that the paper is unnecessary short, in particular with respect to the presentation of the measurement data and a thorough discussion of the limitations of the analysis. This makes it difficult to judge on the significance of the results from this study. Thus I recommend that the paper should only be published after some major modifications.

[Figure]

Major points:

The whole analysis relies on the data presented in Fig. 1. Unfortunately, no information is provided about the original data (number of flights, flight patterns, time series) and no reference is given for the individual flights. Figure 1 shows only median values, with no information on atmospheric variability. I consider it essential that more information is given on the input data, at least please provide flight patterns and information on the variability.

The analysis itself relies on a quasi-Lagrangian approach assuming a spatial and temporal connection between the emission source (Denver), transport and photochemical processing of the plume. Using a statistical analysis (median over several flights?) of measurements at various distances from the Denver source, might establish the spatial connection but not necessarily a temporal relation. I think it is fair to assume, that NOx emissions (and e.g. CO emissions) will exhibit a significant diurnal variation, so that any analysis of the plume evolution has to make sure that this temporal variability of the source has been taken into account. Unfortunately, this question is not addressed in the manuscript. Can we assume that a measurement at a distance of e.g. 50 km away from Denver is indeed related to the NOx emissions at the source 4 hours previously? If not, the limitations of a statistical approach with respect of NOx diurnal variations should be discussed.

Equation 1 relies on the assumption that secondary production of CO can be neglected and I am not sure that this assumption is justified. It can be expected that e.g. HCHO emissions due to mobile sources will lead to enhanced mixing ratios of several ppbv, that will produce additional CO within a time period of several hours. This source might be of similar magnitude as the photochemical sink of CO (1 – 1.5 ppbv according to the estimate made in line 23 on page 4). There might be other secondary sources of CO that could contribute. In any case the limitations and consequences of neglecting secondary CO production on the dilution studies should be discussed.

The discussion on page 6, line 18 – 20 dealing with the derivation of the NOx lifetime from a correlation between NOx and its oxidation products is too short. Please show a least a correlation plot of the data used for this analysis.

More details about the production of peroxy nitrates (LaFranchi et al., 2009) should be provided: e.g. which oxygenated VOC were taken into account?

Minor points:

Equation 2: Is it justified to neglect photochemical sinks for HNO3, e.g. reaction with OH?

Equation 3: Is it justified to neglect physical removal processes (dry deposition, particle production) in the summation of the PNs sinks?